# CO_2_ Storage Monitoring via Time-Lapse Full Waveform Inversion with Automatic Differentiation

**DOI:** 10.3390/nano14020138

**Published:** 2024-01-07

**Authors:** Jixin Yang, Pengliang Yu, Suran Wang, Zheng Sun

**Affiliations:** 1Department of Geoscience, Pennsylvania State University, University Park, PA 16802, USA; 2University of Chinese Academy of Sciences, Beijing 100049, China; 3CNOOC Research Institute Co., Ltd., Beijing 100028, China; wangsr@cnooc.com.cn; 4CUMT-UCASAL Joint Research Center for Biomining and Soil Ecological Restoration, State Key Laboratory of Coal Resources and Safe Mining, China University of Mining and Technology, Xuzhou 221116, China; sunzheng@cumt.edu.cn

**Keywords:** automatic differentiation, CO_2_ capture utilization and storage, time-lapse monitoring, full waveform inversion, deep learning tool

## Abstract

In the field of CO_2_ capture utilization and storage (CCUS), recent advancements in active-source monitoring have significantly enhanced the capabilities of time-lapse acoustical imaging, facilitating continuous capture of detailed physical parameter images from acoustic signals. Central to these advancements is time-lapse full waveform inversion (TLFWI), which is increasingly recognized for its ability to extract high-resolution images from active-source datasets. However, conventional TLFWI methodologies, which are reliant on gradient optimization, face a significant challenge due to the need for complex, explicit formulation of the physical model gradient relative to the misfit function between observed and predicted data over time. Addressing this limitation, our study introduces automatic differentiation (AD) into the TLFWI process, utilizing deep learning frameworks such as PyTorch to automate gradient calculation using the chain rule. This novel approach, AD-TLFWI, not only streamlines the inversion of time-lapse images for CO_2_ monitoring but also tackles the issue of local minima commonly encountered in deep learning optimizers. The effectiveness of AD-TLFWI was validated using a realistic model from the Frio-II CO_2_ injection site, where it successfully produced high-resolution images that demonstrate significant changes in velocity due to CO_2_ injection. This advancement in TLFWI methodology, underpinned by the integration of AD, represents a pivotal development in active-source monitoring systems, enhancing information extraction capabilities and providing potential solutions to complex multiphysics monitoring challenges.

## 1. Introduction

Geological CO_2_ storage stands as an indispensable strategy in addressing the urgent imperative to reduce CO_2_ emissions [1]. Within this domain, CO_2_ capture utilization and storage (CCUS) is a crucial tool, where monitoring the subsurface distribution of CO_2_ is critical; such monitoring serves as a vital tool for refining subsurface physical flow models and ensuring environmental safety by monitoring potential CO_2_ leakages [2,3,4]. Demonstrating its efficacy, time-lapse acoustic monitoring has been identified as a cost-effective technique for subsurface examination, particularly in enhancing oil recovery and sequestration [4,5,6]. For instance, the application of time-lapse traveltime tomography and finite frequency tomography methods in laboratory CO_2_ injection monitoring is notable [7,8,9]. Specifically, time-lapse crosshole traveltime tomography of the Frio CO_2_ plume revealed acoustic velocity decreases of up to 500 m/s, attributable to CO_2_ injection into the brine reservoir [6]. While providing valuable insights, time-lapse traveltime tomography offers limited spatial resolution compared to time-lapse full waveform inversion (TLFWI) [9,10]. Traditional TLFWI, reliant on adjoint gradient methodology, necessitates a precise and often complex gradient formulation. To bypass these complexities, automatic differentiation has been innovatively incorporated into the full waveform inversion workflow [11,12,13].

Compared to traveltime tomography, TLFWI has the capability to refine high-resolution physical images from time-lapse datasets by fully utilizing the physical information embedded in waveform data [14]. The fundamental goal of traditional TLFWI is to reduce the discrepancy between the observed data and their predicted counterpart; typically, this objective is attained via optimization techniques that are gradient-based. Such methods require calculating the gradient of the misfit function in relation to the physical parameters of interest [15,16], a process frequently facilitated by the use of the adjoint-state method. Yet, its derivation and implementation can present challenges across different physical systems or equations [17,18,19]. Presently, there is a lack of generalized frameworks for TLFWI in CCUS applications that can estimate physical parameters without the need for specific gradient derivation in each case. Notably, the monitoring systems in CCUS applications are evolving into multiphysics processes, such as geothermal monitoring, necessitating the use of diverse equations for time-lapse inversion. Therefore, a general framework that obviates the need for case-specific gradient derivation and implementation is highly desirable [12]. This general framework can be applied to seismic inversion problems.

Automatic differentiation (AD), capable of autonomously generating gradients from the computational graph of forward modeling [20], presents a viable alternative to the conventional method. In this process, the computational graph meticulously records the dependencies of arithmetic operations within the forward model and subsequently computes gradients via the chain rule [12]. With the remarkable progress in deep learning, AD has been effectively incorporated into the toolsets of several prominent deep learning computation frameworks, such as PyTorch and TensorFlow [5]. Utilizing the potential of AD, its integration into full waveform inversion (FWI) was recently actualized, fostering a more adaptable and multifaceted approach to waveform inversion [12]. However, the adoption of AD in TLFWI specifically for CO_2_ monitoring workflows remains an underexplored area.

In this research, TLFWI was implemented using AD for applications in CCUS, specifically focusing on CO_2_ monitoring within the Frio-II model framework. Our findings indicate that AD provides results comparable to those obtained through the adjoint-state method. One notable benefit of our methodology is the incorporation of TLFWI into the PyTorch framework. This integration stands in contrast to the traditional reliance on the adjoint-state method, which necessitates the derivation and implementation of distinct gradients for every equation or misfit function. This methodology allows for the adaptation to different inverse problems with minimal alterations in the gradient generation process, potentially enhancing the efficiency and applicability of TLFWI in geophysical studies.

## 2. Methods

### 2.1. Time-Lapse Full Waveform Inverison

The wave equation’s finite difference solution can effectively simulate the waveform propagation in acoustic media [10] given by
(1)$$\frac{\partial^{2}p}{\partial t^{2}}=\nabla \cdot \left(c^{2}\nabla p\right)+f,$$
where p is acoustic displacement, f is the Ricker wavelet source term, and c is the velocity of the computational domain. The interested inversion parameter is c. Recording the dataset from the simulated wavefield p at the receiver location, the predicted data d can be obtained. In the conventional FWI, the $l_{2}$ norm objective function [12] is employed to measure the distance between the observed and predicted dataset, which can be given as
(2)$$E=\frac{1}{2}{\left|d-p\left(m\right)\right|}^{2},$$
where m is the model parameter needs to be reconstructed. In this article, acoustic velocity is considered.

Traditionally, the adjoint-state method is employed to compute the gradient concerning velocity parameters. This approach often includes utilizing an optimizer to ascertain the search direction and step lengths, facilitating the retrieval of model parameters. In contrast, our study adopts AD in PyTorch for gradient computation. This approach circumvents the need for deriving gradients on a case-by-case basis.

Once the gradient is obtained via AD, an optimizer is essential to minimize the misfit function. Instead of opting for commonly used optimizers in deep learning contexts like Adam (Adaptive Moment Estimation) within PyTorch, our approach integrates the preconditioned L-BFGS-B (PLBFGSB) algorithm into our TLFWI. With an initial model parameter set, the PLBFGSB algorithm iteratively searches for gradients, providing a robust mathematical framework [12] to effectively address the challenges in FWI, written as
(3)$$m_{i+1}=m_{i}+\alpha_{i}g_{i},$$
where $\alpha_{i}$ represents the step length, and $g_{i}$ signifies the gradient of the misfit function as outlined in Equation (2). This gradient, related to the velocity parameter, is efficiently calculated through the computational graph of automatic differentiation. Utilizing this framework, FWI iteratively updates the model parameter as per Equation (3), continuing this process until it reaches the convergence minimum.

The main goal of TLFWI is to monitor variations in velocity over a period. This procedure usually entails performing several FWIs on the baseline and subsequent monitoring datasets. Time-lapse images are then produced through TLFWI by deducting the baseline velocity model from the inverted results of the monitoring FWIs. These images provide a dynamic representation of the changes occurring within the velocity parameters over the observed period.

### 2.2. Automatic Differentiation

AD represents an advanced computational mechanism that calculates derivatives through a structured computational graph. This approach is particularly instrumental in deep learning, where it plays a crucial role in training neural networks, primarily by facilitating the backpropagation of loss. AD ensures accurate derivative computations, effectively circumventing the numerical truncation errors often encountered in traditional methods like finite difference calculations. It efficiently manages gradient computations in intricate functions and systems by employing the chain rule, creating a computational graph of elementary differential operators.

Within the scope of employing AD for waveform inversion, the objective function in FWI aligns closely with the loss function typically utilized in deep learning methodologies. This parallel is illustrated in Figure 1. In such a scenario, the velocity model is conceptualized as an array of adjustable parameters aimed at reducing the disparity between observed and simulated acoustic data. The forward modeling process, which utilizes the wave equation, resembles the operation of a neural network through its sequential application of both linear and nonlinear transformations. Our methodology involves specifying parameters like velocity and the positions of sources and receivers to predict acoustic data. The differences in these parameters are automatically computed to derive gradients, which are then employed by a gradient-searching optimizer in a manner akin to training a neural network.

In the application of AD for waveform inversion, a key component is the sophisticated calculation of gradients. The process begins with the specification of parameters such as velocity, source, and receiver locations in the forward modeling phase. This phase involves simulating wave propagation through the subsurface using the wave equation, which yields predicted acoustic data.

Upon generating these predicted data, the core functionality of AD comes into play. AD works by constructing a computational graph where each node represents an operation and each edge represents a dependency between these operations. In the context of FWI, these operations include the various transformations applied during forward modeling—both linear and nonlinear.

The gradient calculation with AD proceeds in two main stages: (1) Forward pass: In this stage, the computational graph is evaluated from the input parameters (like velocity) to the output (predicted data). This step is akin to the propagation of waves in the physical model and is used to compute the discrepancy between the observed and the predicted data. (2) Backward pass (backpropagation): This is where the actual computation of gradients occurs. Starting from the output (the discrepancy between observed and predicted data), AD applies the chain rule of calculus in reverse order through the computational graph. At each node, the partial derivative of the node’s operation is computed with respect to its inputs. This process accumulates the gradients of each parameter with respect to the final discrepancy measure.

The resulting gradients represent how much each parameter needs to change to reduce the discrepancy between the observed and predicted data. This precise gradient calculation is pivotal in AD, as it avoids the truncation errors commonly found in finite difference methods and provides a more accurate direction for updating the parameters.

Finally, these computed gradients are utilized by a gradient-searching optimizer. The optimizer uses these gradients to update the parameters following an optimization process similar to that in neural network training. Through iterative adjustments to the velocity model and other related parameters, this approach seeks to diminish the discrepancy between observed and predicted data, ultimately refining the precision of the waveform inversion.

## 3. Results

### 3.1. Frio-II CO_2_ Storage Model

#### Site Background

The Frio-II CO_2_ pilot project, a landmark in the field of geological CO_2_ sequestration, involves the injection of supercritical CO_2_ into the Blue Sand reservoir. This highly permeable formation is situated at a depth of 1650 m in southern Texas. Our research developed a comprehensive 2D baseline reference velocity model, incorporating intricate local structures identified from borehole logs and gamma ray correlations within the Frio injection borehole (refer to Figure 2a). Within this reservoir, seismic velocities were observed to vary between 2650 m/s and 2675 m/s, in contrast to a uniform velocity of 2700 m/s in the surrounding strata. Figure 2b demonstrates the modeled decrease in seismic velocity, a result of five days of CO_2_ injection at the site, a phenomenon accurately predicted by rock physics models that integrate seismic velocity with saturation levels from a 3D multiphase flow model. A series of 41 time-lapse seismic models, charted at three-hour intervals, was developed. Figure 3a–c depict three representative models, illustrating the P-wave velocity changes at 3, 60, and 120 h postinjection.

A total of 41 time-lapse seismic monitoring datasets were generated through a finely tuned crosshole observation system, meticulously aligned with corresponding time-lapse models. In this detailed arrangement, 32 Ricker sources were strategically placed in the injection borehole and 140 receivers in the observation borehole. This setup was deliberately crafted to enhance the capture and visualization of CO_2_ plume movements in the subsurface. The selection of Ricker wavelets, characterized by a dominant frequency of 120 Hz, was intentional for their efficacy in penetrating geological layers and yielding distinct seismic signals crucial for CO_2_ plume tracking. During each simulation, 32 shots of monitored acoustic data were numerically modeled. The numerical modeling involved solving Equation (1), a process integral to accurately representing the subsurface wave propagation dynamics. This step was crucial in ensuring that the simulated data closely mimicked real-world scenarios, thus enhancing the reliability of the AD-TLFWI method in practical applications. Following these simulations, AD-TLFWI was applied to these datasets, yielding time-lapse images that dynamically delineate the evolution of CO_2_ plumes. The ability of AD-TLFWI to capture these dynamic changes over time is a testament to its sensitivity and accuracy in detecting even subtle variations in subsurface properties. Three exemplary time-lapse velocity results from these models, as illustrated in Figure 4a–c, showcase the detailed evolution of CO_2_ plumes. These images are not just visual representations but carry significant geophysical data that can be analyzed to understand CO_2_ behavior in subsurface environments.

The analysis comparing time-lapse seismic data, as presented in Figure 3 and Figure 4, particularly focusing on the high-resolution imaging post-CO_2_ injection, underscores the proficiency of the AD-TLFWI method in discerning velocity variations accurately. The alignment of these detected changes with the actual ground truth data affirms the method’s precision in mirroring key velocity alterations occurring during the injection phase. This precision is further evidenced by the resolution’s close correlation with real subsurface shifts. The results, depicting the spatiotemporal fluctuations within the injected CO_2_, provide crucial insights into the dynamics governing CO_2_ behavior in subterranean settings. The oscillations observed in the inverted results point to the inherent challenges in TLFWI. One such challenge is the intrinsic nonlinearity of the FWI process, which can introduce complexities in accurately interpreting seismic data. These nonlinearities are often exacerbated by limitations in the number of iterations conducted during the inversion process. While increasing the number of iterations could theoretically enhance the fidelity and resolution of the time-lapse images, it is crucial to consider the computational trade-offs. Higher iteration counts typically lead to increased computational demands, which can affect the efficiency and practicality of the monitoring project. This is particularly relevant in large-scale or continuous monitoring scenarios where computational resources and time are limiting factors. In advancing AD-TLFWI methodologies, striking an optimal balance between image quality and computational efficiency becomes paramount. The ability of the method to deliver high-resolution images is undeniably valuable for understanding subsurface phenomena, but the computational intensity required to achieve these results must be sustainable. Future advancements in AD-TLFWI could focus on optimizing algorithmic efficiency, perhaps through more sophisticated inversion strategies or leveraging advances in computational hardware, to maintain high-quality imaging while minimizing resource demands. This balance is not only a technical consideration but also impacts the broader applicability and scalability of AD-TLFWI in various geophysical monitoring scenarios, including those beyond CO_2_ sequestration. As such, ongoing research and development in this field is essential to fully harness the potential of AD-TLFWI in environmental monitoring and other geophysical applications.

Figure 5, a culmination of 41 time-lapse seismic studies, effectively showcases the dynamic imaging capabilities of AD-TLFWI in monitoring supercritical CO_2_ movement through the Blue Sand reservoir following injection. These visualizations detail the CO_2_ plume’s dispersion and velocity shifts within the reservoir, initially concentrated around the injection point and gradually expanding and intensifying in response to the reservoir’s varied properties. Notably, the velocity shifts observed, ranging from 0 to a decrease of up to 500 m per second, provide valuable insights into the CO_2_ storage efficiency and containment security. This slowdown in wave propagation, influenced by factors like reservoir heterogeneity and CO_2_ saturation levels, highlights the plume’s interaction with the subsurface environment. The effectiveness of AD-TLFWI, demonstrated here, lies in its enhanced resolution and clarity of time-lapse images, which contributes to a deeper understanding of CO_2_ behavior and facilitates a balance between image clarity and computational resource requirements. This advancement not only optimizes resource allocation in seismic studies but also signals a significant leap in environmental safety measures for CO_2_ sequestration strategies, marking an important development in geophysical monitoring technologies.

Figure 6 showcases the efficacy of the AD-TLFWI approach in correlating the time-lapse seismic data with the simulated models, evident through the concurrence of wave patterns. The comparison in Figure 6d reveals a notable difference in the seismic signatures pre- and post-CO_2_ injection (Figure 6a,b, respectively), which provides a clear visualization of the significant impact of CO_2_ on seismic waveform propagation. The postinjection data demonstrate a noticeable alteration in seismic waves, likely due to changes in the acoustic properties of the subsurface environment influenced by CO_2_ saturation. Figure 6c vividly illustrates the significant correlation in wave propagation patterns between the synthetic and actual observed time-lapse data, underscoring the precision of the AD-TLFWI method. The discrepancies observed in Figure 6f further suggest that the AD-TLFWI is not only replicating but also precisely capturing the temporal and spatial changes in monitoring the subsurface. This comparison is critical to demonstrating the reliability of the predictions and interpretations in CO_2_ sequestration and monitoring. The ability of AD-TLFWI to track these changes is indicative of its robustness in monitoring the geological phenomena. The precision of the proposed method is particularly relevant in detecting subtle changes over elapsed time, which is critical in monitoring the efficiency and safety of CO_2_ storage. The seismic response transformation of post-CO_2_ injection could be indicative of various subsurface changes, including alterations in pore fluid pressure, changes in fluid saturation, or even subtle shifts in rock properties. Integrating these observations with the previous analysis of AD-TLFWI inversion at the Frio-II CO_2_ injection site, it becomes evident that AD-TLFWI is not just a tool for creating high-resolution images but is also instrumental in accurately interpreting the dynamic behavior of CO_2_ in subsurface reservoirs. The use of this method goes beyond simple imaging; it provides a holistic approach for observing, examining, and forecasting the movements of CO_2_ plumes across diverse geological environments.

The results depicted in Figure 6, combined with the comprehensive insights gained throughout this research, highlight the exceptional capability of the AD-TLFWI method as a pioneering instrument in geophysical monitoring. This technique marks a notable leap forward in our understanding and management of CO_2_ sequestration processes, making a significant contribution to environmental geophysics. It stands out as a key tool in the ongoing quest for sustainable and secure CO_2_ storage solutions, offering a new level of precision and efficiency in monitoring techniques. This advancement not only enhances our understanding of subterranean CO_2_ dynamics but also aligns with global efforts to develop environmentally responsible and effective strategies for carbon management.

## 4. Conclusions

The imperative to enhance monitoring techniques for CO_2_ sequestration and storage, pivotal for climate change mitigation, forms the foundation of this study. Recognizing the need for more precise and dynamic methods to track large-scale CO_2_ storage behavior in subsurface environments, our research introduces the integration of AD with TLFWI to refine the distribution and migration of CO_2_.

Methodologically, this research marks a departure from conventional seismic monitoring techniques. By harnessing deep learning frameworks, the AD-TLFWI approach optimizes the gradient computation essential in CO_2_ monitoring, thereby improving the accuracy and adaptability of subsurface imaging in various geophysical settings. The application of this method at the Frio-II CO_2_ injection site has been instrumental in producing high-resolution time-lapse images, crucial for monitoring the subsurface migration and distribution of CO_2_ plumes.

In the application of AD-TLFWI at the Frio-II site, the key point lies in its ability to eliminate the need for deriving gradient formulas traditionally required in FWI by leveraging automatic differentiation for gradient computation. This significant advancement streamlines the inversion process, enhancing both efficiency and precision. Furthermore, AD-TLFWI has proven highly effective in imaging the location, distribution, and migration of CO_2_ within subsurface storage, marking it as a tool for large-scale, long-term monitoring of CO_2_ sequestration. These findings are crucial for confirming the effectiveness and safety of CO_2_ sequestration strategies, offering insights for optimizing sequestration methods. However, this study also acknowledges limitations, particularly the AD-TLFWI’s sensitivity to environmental noise and proposed method is to invert the single physical parameter, which may affect the accuracy of the inversion results. Addressing these challenges in future research will be key to enhancing the misfit function and multiparameter inversion strategy of AD-TLFWI.

Looking forward, the research aims to refine the robustness of the AD-TLFWI framework, especially in challenging geophysical environments. The potential broadening of its application across different areas of geophysical monitoring, including renewable energy exploration and environmental protection, opens new avenues for research and development. This study lays a solid foundation for future advancements in geophysical monitoring methodologies, contributing significantly to the field of CCUS and aligning with global efforts towards sustainable energy and environmental stewardship.

## Figures and Tables

**Figure 1 nanomaterials-14-00138-f001:**
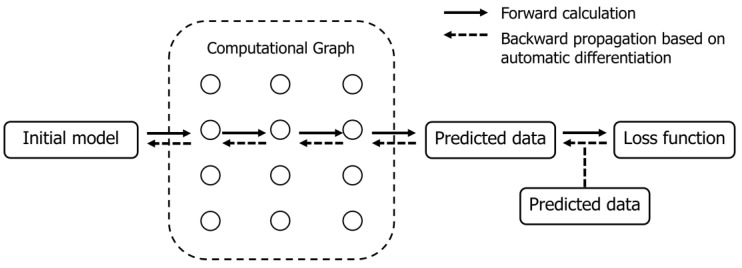
Workflow of AD-TLFWI with AD framework.

**Figure 2 nanomaterials-14-00138-f002:**
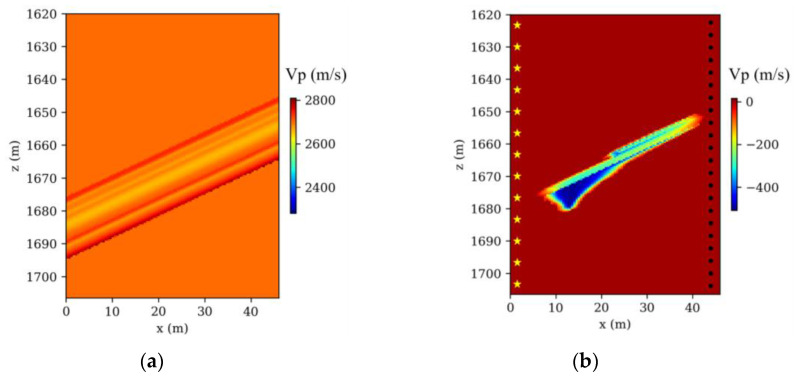
Frio-II CO_2_ model. (**a**) Baseline velocity model; (**b**) time-lapse velocity change after 120 h CO_2_ injection (yellow stars represent source and black dots stand for receivers).

**Figure 3 nanomaterials-14-00138-f003:**
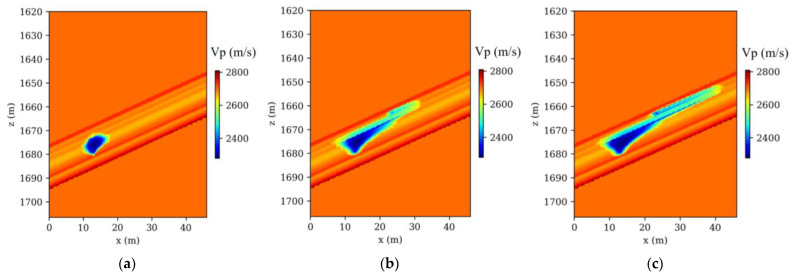
True time-lapse velocity models: (**a**) at 3 h; (**b**) at 60 h; (**c**) at 120 h, respectively.

**Figure 4 nanomaterials-14-00138-f004:**
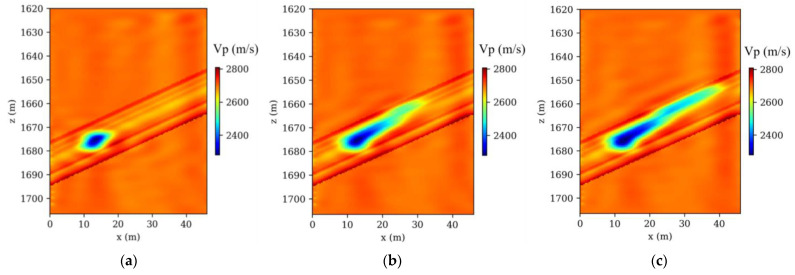
Inverted time-lapse velocity models: (**a**) at 3 h; (**b**) at 60 h; (**c**) at 120 h, respectively.

**Figure 5 nanomaterials-14-00138-f005:**
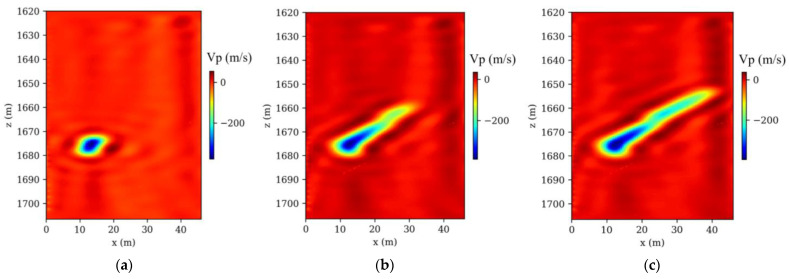
Time-lapse velocity changes: (**a**) at 3 h; (**b**) at 60 h; (**c**) at 120 h, respectively.

**Figure 6 nanomaterials-14-00138-f006:**
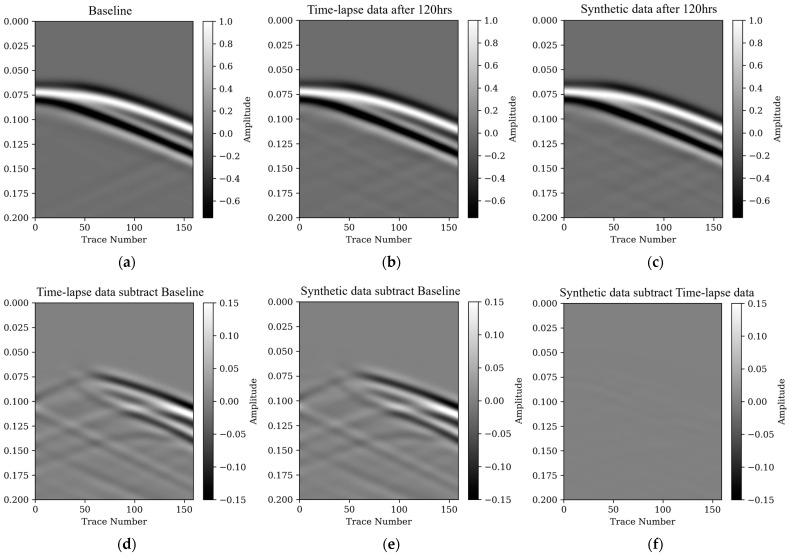
Waveform comparison: (**a**) waveform simulated using baseline velocity; (**b**) waveform simulated using true time-lapse velocity after 120 h injection; (**c**) waveform simulated using the inverted velocity after 120 h injection; (**d**) data residual by subtracting waveform (**a**) from waveform (**b**); (**e**) data residual by subtracting waveform (**a**) from waveform (**c**); (**f**) data residual by subtracting waveform (**b**) from waveform (**c**).

## Data Availability

The data presented in this study are not available due to privacy.

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
