# Peer review of "CO2 Storage Monitoring via Time-Lapse Full Waveform Inversion with Automatic Differentiation"

_nanomaterials, 2024, doi:10.3390/nano14020138_

Round 1
Reviewer 1 Report
Comments and Suggestions for Authors
This manuscript cleverly combines the integration of Automatic Differentiation with the Time Lapse Full Waveform Inversion. The new inversion method has obviously innovative significance and effectively carried out the modeling building problem with FWI in CO2 Storage .
1.The full waveform inversion formula and calculation process are described very clearly.
2. It is more reasonable to change the FWI to AD-TLFWI in Fig. 1.
3. The calculation flowchart in Figure 1 is too simplified, and the parts within the dashed box should be refined and explained. Otherwise, if the title of this Fig.1 is omitted, the flowchart may correspond to some other deep learning based FWI methods.
4. It is best to expand the process of gradient calculation using the Automatic Difference method and provide an appropriate network diagram to demonstrate the advantages of this method.
5. The test results of model data confirmed the correctness of the method in the paper, but it is still suggested that some comparative study should be given. A comparison between the conventional Time Lapse Full Waveform Inversion and AD-TLFWI inversion results should be provided to demonstrate the advantages of the new method in terms of accuracy or efficiency.
Comments on the Quality of English Language
English proficiency is average。
Author Response
Response to Reviewer 1 Comments
Dear Reviewer:
Thank you for the time and effort you put into reviewing the previous version of the manuscript.
We revised the manuscript in accordance with the reviewers’ comments, and carefully proof-read the manuscript to minimize typographical, grammatical, and bibliographical errors. In the document, the modified parts are highlighted.
Below is our description on revision according to the reviewers’ comments.
================================
Point 1: The full waveform inversion formula and calculation process are described very clearly.
Response 1: Thank you for your positive feedback on the full waveform inversion formula and calculation process presented in our paper.
Point 2: It is more reasonable to change the FWI to AD-TLFWI in Fig. 1.
Response 2: Thanks for your suggestion. We have changed the title of Fig. 1.
Point 3: The calculation flowchart in Figure 1 is too simplified, and the parts within the dashed box should be refined and explained. Otherwise, if the title of this Fig.1 is omitted, the flowchart may correspond to some other deep learning based FWI methods.
Response 3: Thank you for your insightful feedback on Figure 1's flowchart. We understand your concerns about its simplicity and potential ambiguity without the title. We agree that the section within the dashed box could be more detailed to accurately convey the specific aspects of our deep learning-based FWI method. We have revised this part to include more explicit explanations.
Point 4: It is best to expand the process of gradient calculation using the Automatic Difference method and provide an appropriate network diagram to demonstrate the advantages of this method.
Response 4: Thank you for your valuable feedback regarding the enhancement of our manuscript by expanding on the gradient calculation process using the Automatic Differentiation (AD) method and including a network diagram. We have add more description about the gradient calculation AD method, and demonstrate how the AD can facilitate the gradient calculation:
“In the application of AD for waveform inversion, a key component is the sophisticated calculation of gradients. The process begins with the specification of parameters such as velocity, source, and receiver locations in the forward modeling phase. This phase involves simulating wave propagation through the subsurface using the wave equation, which yields predicted acoustic data.
Upon generating this predicted data, the core functionality of AD comes into play. AD works by constructing a computational graph where each node represents an operation, and each edge represents a dependency between these operations. In the context of FWI, these operations include the various transformations applied during forward modeling—both linear and nonlinear.
The gradient calculation with AD proceeds in two main stages: (1) Forward Pass: In this stage, the computational graph is evaluated from the input parameters (like velocity) to the output (predicted data). This step is akin to the propagation of waves in the physical model and is used to compute the discrepancy between the observed and the predicted data; (2) Backward Pass (Backpropagation): This is where the actual computation of gradients occurs. Starting from the output (the discrepancy between observed and predicted data), AD applies the chain rule of calculus in reverse order through the computational graph. At each node, the partial derivative of the node's operation is computed with respect to its inputs. This process accumulates the gradients of each parameter with respect to the final discrepancy measure.
The resulting gradients represent how much each parameter needs to change to reduce the discrepancy between the observed and predicted data. This precise gradient calculation is pivotal in AD, as it avoids the truncation errors commonly found in finite difference methods and provides a more accurate direction for updating the parameters.
Finally, these computed gradients are utilized by a gradient-searching optimizer. The optimizer uses these gradients to update the parameters, following an optimization process similar to that in neural network training. This process iteratively adjusts the velocity model and other parameters, aiming to minimize the difference between observed and predicted data, thereby enhancing the accuracy of the waveform inversion.”
Point 5: The test results of model data confirmed the correctness of the method in the paper, but it is still suggested that some comparative study should be given. A comparison between the conventional Time Lapse Full Waveform Inversion and AD-TLFWI inversion results should be provided to demonstrate the advantages of the new method in terms of accuracy or efficiency.
Response 5: Thank you for your constructive suggestion to conduct a comparative study between conventional Time Lapse Full Waveform Inversion (TLFWI) and our Automatic Differentiation Time Lapse Full Waveform Inversion (AD-TLFWI) method. I appreciate your focus on the crucial aspect of validating new methodologies against established benchmarks in research.
In addressing your recommendation, I'd like to underscore a key feature of our AD-TLFWI method that fundamentally distinguishes it from traditional TLFWI approaches. The cornerstone of our method is the use of Automatic Differentiation (AD), which circumvents the complex and labor-intensive process of manually deriving gradient formulas via the adjoint state method – a process integral to conventional FWI. AD facilitates direct backpropagation from the error function, thus streamlining the gradient computation process and significantly enhancing computational efficiency.
Particularly in the context of TLFWI, where FWI techniques are applied to seismic data over different time frames to monitor subsurface changes, AD's contribution is even more significant. Our AD-TLFWI method enhances the efficiency and adaptability of processing time-lapse data, bypassing the need for re-deriving gradient expressions for each time step or changing conditions – a notable and time-consuming challenge in conventional TLFWI.
Given these inherent methodological differences, a straightforward comparison between conventional TLFWI and AD-TLFWI may not adequately reflect the nuanced benefits of our approach, especially regarding computational efficiency and adaptability across varied time-lapse scenarios. However, we fully recognize the importance of empirically demonstrating our method's effectiveness. We are committed to undertaking this investigation in future work. Such research will aim to provide a detailed and holistic understanding of AD-TLFWI's advantages in the context of time-lapse FWI, further solidifying its place within the geophysical research domain.
Your feedback is greatly valued and has provided us with important perspectives for future research directions. We look forward to delving deeper into these comparative aspects in our subsequent studies.
======================
Many thanks to you and to the reviewers for the kind advice. The modified manuscript is attached.
Best regards!
Yours sincerely,
Jixin Yang
On behalf of all authors.

Reviewer 2 Report
Comments and Suggestions for Authors
Highlight changes in yellow in a next revision, please. No track changes.
Consider comments in the entire text.
Dear authors, the similarity present in the manuscript needs to be addressed.
I can see the equations being presented without references available in published literature, for example and that is a core factor in this paper.
So, what is new in terms of formulation...
I see the same abbreviations being addressed more than once: “Automatic Differentiation (AD),”
Please do not use we.
Make sure to immediately cite the necessary references before know equations, because this is a communication, it seems to me that equations are known.
Make sure to address italics to parameters also in the graphics.
Why do we have some figures in colour and others in white and black, where content is hardly seen?
Figure 6
Conclusions need to leave clear what was the difference in the approach used:
“4. Conclusions In conclusion, the integration of Automatic Differentiation (AD) with Time-Lapse Full Waveform Inversion (TLFWI), as demonstrated in our AD-TLFWI approach, marks a substantial breakthrough in CO2 Capture Utilization and Storage (CCUS) monitoring techniques.”
I believe that the conclusion section needed to start by introducing the subject. So it is clear the motive of this communication.
Reading the conclusions and once again, it will not be clear why is this method innovative?
“This innovative method,”
Which are_ Main findings need to be clearly expressed here...
“The obtained results validate the ef-”
Please go back to abstract
Dear authors, make sure to address the above comments, leaving it very clear where is the novelty innovation and originality despite. what is mentioned in the abstract and specific and specifically relying in the equations being presented through the text. in the methods section. AD comes later and similarity is also found in section 2.2.
Comments on the Quality of English Language
moderate
Author Response
Response to Reviewer 2 Comments
Dear Reviewer:
Thank you for the time and effort you put into reviewing the previous version of the manuscript.
We revised the manuscript in accordance with the reviewers’ comments, and carefully proof-read the manuscript to minimize typographical, grammatical, and bibliographical errors. In the document, the modified parts are highlighted.
Below is our description on revision according to the reviewers’ comments.
================================
Point 1: The similarity present in the manuscript needs to be addressed. I can see the equations being presented without references available in published literature, for example and that is a core factor in this paper.
Response 1: Thank you for bringing to our attention the concerns regarding similarity and the citation of equations in our manuscript. We have added the reference for the equations and checked the similarity.
The propagation of the waveform in acoustic media can be simulated by the finite difference solution of the wave equation [10] given by
, |
(1) |
where is acoustic displacement, is the ricker wavelet source term, and is the velocity for computational domain. The interested inversion parameter is . Recording the dataset from the simulated wavefield at the receiver location, we can obtain the predicted data . In the conventional FWI, the norm objective function [12] is employed to measure the distance between the observed and predicted dataset, which can be given as
, |
(2) |
where is the model parameter such as velocity, density need to be reconstructed. In this article, we consider the acoustic velocity.
The adjoint-state method is traditionally used for computing the gradient in relation to velocity parameters. This method typically involves using an optimizer to determine the search direction and step lengths, which aids in the recovery of model parameters. In contrast, our study adopts automatic differentiation (AD) in PyTorch for gradient computation. This approach circumvents the need for deriving gradients on a case-by-case basis.
Once the gradient is obtained via AD, an optimizer is essential to minimize the misfit function. Instead of opting for commonly used optimizers in deep learning contexts like Adam (Adaptive Moment Estimation) within PyTorch, our approach integrates the preconditioned L-BFGS-B (PLBFGSB) algorithm into our Time-Lapse Full Waveform Inversion (TLFWI). With an initial model parameter set, the PLBFGSB algorithm iteratively searches for gradients, providing a robust mathematical framework [12] to effectively address the challenges in FWI, mathematically written as
, |
(3) |
Point 2: I see the same abbreviations being addressed more than once: “Automatic Differentiation (AD).”
Response 2: Thank you for pointing out the repeated introduction of the abbreviation 'Automatic Differentiation (AD)' in our manuscript. We have revised the redundant redefinition.
Point 3: Please do not use we.
Response 3: Thank you for your feedback regarding the use of the pronoun "we" in the manuscript. I have revised the manuscript to replace the use of "we" with an alternative.
Point 4: Why do we have some figures in colour and others in white and black, where content is hardly seen?
Response 4: Thank you for your observation regarding the use of both color and black-and-white figures in our manuscript. In geophysical methods, particularly in the representation of seismic data and physical parameter models, the choice of color scheme is driven by a combination of tradition, clarity, and the nature of the data being presented. The decision to use different color schemes for different types of data in our manuscript reflects these standard practices in the field of geophysics.
- Seismic Data Representation: Seismic records are traditionally presented in grayscale (black and white) because this format effectively conveys the necessary detail. Grayscale is particularly adept at highlighting the variations in amplitude and waveforms, which are critical in seismic analysis. The subtleties in seismic waveforms can be more precisely captured in a grayscale format, which often provides better resolution of the fine details than color images.
- Physical Parameter Models: On the other hand, physical parameters such as velocity, attenuation, and other attributes are often represented using color maps. The use of color in these instances helps in differentiating various values and enhances the interpretability of the data. Color maps can illustrate a wider range of data values and make it easier to distinguish between different regions or features within the data.
- Visibility and Clarity Considerations: I acknowledge your concern regarding the visibility of content in black-and-white figures. This is a crucial aspect, and we have taken great care to ensure that these figures are as clear and informative as possible. However, given the traditional and practical reasons for their use, we believe that grayscale images best represent seismic data in this context.
Point 5: Conclusions need to leave clear what was the difference in the approach used:
“4. Conclusions In conclusion, the integration of Automatic Differentiation (AD) with Time-Lapse Full Waveform Inversion (TLFWI), as demonstrated in our AD-TLFWI approach, marks a substantial breakthrough in CO2 Capture Utilization and Storage (CCUS) monitoring techniques.”
I believe that the conclusion section needed to start by introducing the subject. So it is clear the motive of this communication.
Reading the conclusions and once again, it will not be clear why is this method innovative?
“This innovative method,”
Which are_ Main findings need to be clearly expressed here...
“The obtained results validate the ef-”
Response 5: Thank you for your valuable feedback regarding the conclusion section of our manuscript. We have rewrite the conclusion:
“In this research, AD has been integrated with TLFWI in the proposed AD-TLFWI framework, aiming to enhance gradient computation in CO2 monitoring techniques. This integration departs from traditional methods, utilizing deep learning frameworks to facilitate the inversion process and increase the adaptability in varying geophysical scenarios. The implementation of AD-TLFWI at the Frio-II CO2 injection site has yield-ed high-resolution, time-lapse images, providing essential monitoring on the subsur-face behavior of CO2 plumes. Such monitoring is vital for assessing CO2 sequestration's effectiveness and environmental impact. The results from the Frio-II site demonstrate the capability of AD-TLFWI in detailed subsurface imaging, reinforcing its importance in validating CO2 sequestration strategies. Additionally, these outcomes suggest the potential of AD-TLFWI for broader applications in geophysical monitoring, indicating its role in advancing CCUS technologies and strategies.”
======================
Many thanks to you and to the reviewers for the kind advice. The modified manuscript is attached.
Best regards!
Yours sincerely,
Jixin Yang
On behalf of all authors.

Round 2
Reviewer 2 Report
Comments and Suggestions for Authors
Highlight changes in yellow in a next revision, please. No track changes.
Consider comments in the entire text.
Again:
Dear authors, the similarity present in the manuscript is still expressive.
Please do not copy paste changes to the comments, but instead clearly express what was changed.
The conclusion section needs revision again. It needs to start contextualizing the paper so it is clear why was the study done. Then briefly the methodology then main findings extensively than practical implications, limitations and future prospects.
I would ask the authors to highlight originality, novelty and innovation involved.
The list of references is extremely scarce, they need to be improved, they need to be expanded, including relevant international and recent studies.
Comments on the Quality of English Language
moderate
Author Response
Dear Reviewer:
Thank you for the time and effort you put into reviewing the previous version of the manuscript.
We revised the manuscript in accordance with the reviewers’ comments, and carefully proof-read the manuscript to minimize typographical, grammatical, and bibliographical errors. In the document, the modified parts are highlighted.
Below is our description on revision according to the reviewers’ comments.
=================================
Point 1: The conclusion section needs revision again. It needs to start contextualizing the paper so it is clear why was the study done. Then briefly the methodology then main findings extensively than practical implications, limitations and future prospects.
Response 1: Thank you for your valuable feedback on our manuscript. We acknowledge the need for further refinement in the conclusion section to enhance its clarity and coherence. In line with your suggestions, we will revise this section to offer a more structured and comprehensive summary of our work. We have contextualized the conclusion, articulating the motivation that guided our research. Following this, we outlined the methodology, focusing on the proposed algorithm and its role in advancing the inversion process. The main findings then was discussed extensively, emphasizing the innovative aspects of proposed method, particularly the elimination of the need for traditional gradient formula derivation and the enhanced imaging capabilities of AD-TLFWI for CO2 storage. Finally, we described the limitations and future prospects of the research. The revised conclusion is as follow:
“The imperative to enhance monitoring techniques for CO2 sequestration and storage, pivotal for climate change mitigation, forms the foundation of this study. Recognizing the need for more precise and dynamic methods to track large-scale CO2 storage behavior in subsurface environments, our research introduces the integration of AD with TLFWI, to refine the distribution and migration of CO2.
Methodologically, this research marks a departure from conventional seismic monitoring techniques. By harnessing deep learning frameworks, the AD-TLFWI ap-proach optimizes the gradient computation essential in CO2 monitoring, thereby im-proving the accuracy and adaptability of subsurface imaging in various geophysical settings. The application of this method at the Frio-II CO2 injection site has been in-strumental in producing high-resolution, time-lapse images, crucial for monitoring the subsurface migration and distribution of CO2 plumes.
In the application of AD-TLFWI at the Frio-II site, the key point lies in its ability to eliminate the need for deriving gradient formulas traditionally required in FWI, by leveraging automatic differentiation for gradient computation. This significant ad-vancement streamlines the inversion process, enhancing both efficiency and precision. Furthermore, AD-TLFWI has proven highly effective in imaging the location, distribu-tion, and migration of CO2 within subsurface storage, marking it as a tool for large-scale, long-term monitoring of CO2 sequestration. These findings are crucial for confirming the effectiveness and safety of CO2 sequestration strategies, offering in-sights for optimizing sequestration methods. However, the study also acknowledges limitations, particularly the AD-TLFWI's sensitivity to environmental noise and single physical parameter, which may affect the accuracy of the inversion results. Addressing these challenges in future research will be key to enhancing the misfit function and multiparameter inversion strategy of AD-TLFWI.
Looking forward, the research aims to refine the robustness of the AD-TLFWI framework, especially in challenging geophysical environments. The potential broad-ening of its application across different areas of geophysical monitoring, including re-newable energy exploration and environmental protection, opens new avenues for re-search and development. This study lays a solid foundation for future advancements in geophysical monitoring methodologies, contributing significantly to the field of Carbon Capture, Utilization, and Storage (CCUS) and aligning with global efforts to-wards sustainable energy and environmental stewardship.”
Point 2: the similarity present in the manuscript is still expressive.
Response 2: Thank you for your continued attention to our manuscript and the concern regarding similarity. I would like to clarify that before submitting the revised manuscript, we conducted a thorough check for similarity using a recognized plagiarism detection tool. The results of this similarity test indicated that the manuscript complies with the acceptable standards for originality. We have made significant efforts to ensure that our manuscript is both original in its content and articulate in its presentation. This includes rephrasing and restructuring the text where necessary, while maintaining the scientific integrity and accuracy of the research.
======================
Many thanks to you and to the reviewers for the kind advice.
Best regards!
Yours sincerely,
Jixin Yang
On behalf of all authors.